applied mathematics/differential equations/ mathematical physics

Darboux transformation, SIdV equation, Korteweg–de Vries equation, kink solution, decomposition, phase shift

**Author for correspondence:**
Jingsong He
e-mail: hejingsong@szu.edu.cn; hejingsong@nbu.edu.cn

# Kink-type solutions of the SIdV equation and their properties

Guofei Zhang[1], Jingsong He[2], Lihong Wang[1] and Dumitru Mihalache[3]

[1]School of Mathematics and Statistics, Ningbo University, Zhejiang 315211, People's Republic of China
[2]Institute for Advanced Study, Shenzhen University, Shenzhen, Guangdong 518060, People's Republic of China
[3]Horia Hulubei National Institute for Physics and Nuclear Engineering, PO Box MG-6, Magurele 077125, Romania

GZ, 0000-0001-5421-351X; JH, 0000-0002-2068-1849

We study the nonlinear integrable equation, $u_t + 2((u_x u_{xx})/u) = \epsilon u_{xxx}$, which is invariant under scaling of dependent variable and was called the SIdV equation (see Sen *et al.* 2012 *Commun. Nonlinear Sci. Numer. Simul.* **17**, 4115–4124 (doi:10.1016/j.cnsns. 2012.03.001)). The order-$n$ kink solution $u^{[n]}$ of the SIdV equation, which is associated with the $n$-soliton solution of the Korteweg–de Vries equation, is constructed by using the $n$-fold Darboux transformation (DT) from zero 'seed' solution. The kink-type solutions generated by the onefold, twofold and threefold DT are obtained analytically. The key features of these kink-type solutions are studied, namely their trajectories, phase shifts after collision and decomposition into separate single kink solitons.

## 1. Introduction

Sen *et al.* [1] casually found the following equation:

$$\hat{u}_t + \left(\frac{2\hat{u}_{xx}}{\hat{u}}\right)\hat{u}_x = \hat{u}_{xxx}, \tag{1.1}$$

which is the simplest member in a vast hierarchy of nonlinear partial differential equations sharing the single soliton solution expressed in a form as $\text{sech}^2$ with the well-known Korteweg–de Vries (KdV) equation

$$\omega_t + 6\omega\omega_x - \omega_{xxx} = 0. \tag{1.2}$$

Equation (1.1) was extended by Sen *et al.* [1] as

$$\tilde{u}_t + 2a\frac{\tilde{u}_x\tilde{u}_{xx}}{\tilde{u}} = \epsilon a\tilde{u}_{xxx}, \quad a \text{ and } \epsilon \text{ are two constants}, \tag{1.3}$$

which is called the SIdV equation because of its scale-invariant property and the above relationship with the KdV equation. Two conservation laws and periodic travelling waves of the SIdV equation were given in [1]. Very recently, Silva *et al.* [2] have put forward the connection between SIdV, Airy (linear KdV), KdV and modified KdV equations. We set $\epsilon a = 1$ and $\epsilon = 2/3$, then equation (1.3) becomes [1,2]

$$u_t + 3\frac{u_x u_{xx}}{u} = u_{xxx}, \tag{1.4}$$

which admits the following Lax pair:

$$L = -D_x^2 + \frac{u_{xx}}{u} \quad \text{and} \quad B = 4D_x^3 - 6\frac{u_{xx}}{u}D_x - 3\left(\frac{u_{xxx}}{u} - \frac{u_x u_{xx}}{u^2}\right). \tag{1.5}$$

Note that the SIdV equation (1.4) was deduced as early as almost 30 years ago in [3] from a point of view of the governing equation of eigenfunction by eliminating the potential function $\omega$ in Lax pair, and this equation was also re-derived in [4] by revealing the direct links between the well-known Sylvester matrix equation and soliton equations.

If $\omega$ is a solution of the KdV equation (1.2), then a solution of the SIdV equation (1.4) can be obtained by solving the following linear system [2]:

$$\left.\begin{array}{l} u_t + 3\omega u_x = u_{xxx} \\ -u_{xx} + \omega u = 0. \end{array}\right\} \tag{1.6}$$

and

It is highly non-trivial to get the solution $u$ of the SIdV equation (1.4) from a known solution $\omega$ of the KdV equation from the above coupled linear system of variable coefficient partial differential equations. Silva *et al.* [2] have obtained a kink-type solution $u$ from a single soliton solution $\omega$ by solving the associated Legendre equation that is reduced from the second formula of equation (1.6). Obviously, the idea to get the solution $u$ from the solution $\omega$ through solving the linear system (1.6) might be not applicable if the known solution $\omega$ is a two-soliton solution or other complicated solution of the KdV equation. Thus, there is an interesting open problem: can we find new solutions $u$ associated with the higher-order solitons $\omega$ of the KdV equation in another way? The purpose of this paper is to provide an affirmative answer to this question by using the Darboux transformation (DT) [5,6] of the KdV equation. We will also study the key properties of interaction of multi-kink solitons of the SIdV equation.

The organization of this paper is as follows. In §2, we clearly illustrate a crucial relationship between the solution $u$ of the SIdV equation and the eigenfunction $\psi$ of the Lax pair of the KdV equation, and then provide the $n$-fold DT of the SIdV equation. In §3, three explicit kink-type solutions of the SIdV equation are constructed by using the DT from zero 'seed' solution. Furthermore, the key characteristics of the kink-type solitons of the SIdV equation are studied, namely the trajectories, the phase shifts after collision, and the decomposition into separate single kinks. The conclusion and discussion of results are given in §4.

## 2. The $n$-fold Darboux transformation of the SIdV equation

The KdV equation admits the following Lax pair:

$$\left.\begin{array}{l} \psi_t = 4\psi_{xxx} - 6\omega\psi_x - 3\omega_x\psi \\ -\psi_{xx} + \omega\psi = \lambda\psi. \end{array}\right\} \tag{2.1}$$

and

If we set $\lambda \to 0$, then

$$\left.\begin{array}{l} \psi_t + 3\omega\psi_x = \psi_{xxx} \\ -\psi_{xx} + \omega\psi = 0. \end{array}\right\} \tag{2.2}$$

and

Comparing the above equations with equation (1.6), it implies a direct relationship between the solutions of the SIdV equation and the eigenfunction $\psi$ of the Lax pair of the KdV equation, namely $u = \psi|_{\lambda=0}$. This observation is crucial for us such that we can solve the SIdV equation by using the DT of the KdV equation. Note that a general equation of the eigenfunction $\psi$ was constructed in [3] by eliminating the potential function $\omega$ in equations of the Lax pair; see eqn (2.3) of Konopelchenko [3].

By setting a 'seed' solution $\omega = 0$, then the corresponding eigenfunctions are

$$\psi = \exp k(x + 4k^2 t), \quad \text{for } \lambda = -k^2 \tag{2.3}$$

and

$$\psi_n = \begin{cases} \cosh k_n(x + 4k_n^2 t), & n = 2j - 1, \\ \sinh k_n(x + 4k_n^2 t), & n = 2j, \end{cases} \quad \text{for } \lambda_n = -k_n^2. \tag{2.4}$$

Here, $k > 0$, $k_n > \cdots > k_2 > k_1 > 0$.

The $n$-fold DT of the KdV equation yields the new solutions [5,6]

$$\omega^{[n]} = -2(\ln W(\psi_1, \ldots, \psi_n))_{xx} \tag{2.5}$$

and

$$\psi^{[n]} = \frac{W(\psi_1, \ldots, \psi_n, \psi)}{W(\psi_1, \ldots, \psi_n)}, \tag{2.6}$$

from a zero 'seed' solution $\omega = 0$. Here

$$W(\psi_1, \psi_2, \ldots, \psi_n) = \begin{vmatrix} \psi_1 & \psi_{1x} & \cdots & \psi_{1x}^{(n-1)} \\ \psi_2 & \psi_{2x} & \cdots & \psi_{2x}^{(n-1)} \\ \vdots & \vdots & \ddots & \vdots \\ \psi_n & \psi_{nx} & \cdots & \psi_{nx}^{(n-1)} \end{vmatrix}$$

is the usual Wronskian determinant of $n$ eigenfunctions $\psi_1$, $\psi_2, \ldots,$ $\psi_n$, and $\psi_{jx}^{(k)}(j = 1, 2, \ldots, n; k = 1, 2, 3, \ldots, n - 1)$ denotes the order-$k$ derivative of $\psi_j$ with respect to $x$.

Hence the above $n$-fold DT produces

$$u^{[n]} = \psi^{[n]}|_{\lambda=0}, \tag{2.7}$$

which is a new solution of the SIdV equation (1.4) associated with an $n$-soliton solution $\omega^{[n]}$ of the KdV equation. Here $u^{[n]}$ in equation (2.7) is a general expression of the order-$n$ kink type solution of the SIdV equation. Unlike the method given in [2], we get $u^{[n]}$ associated with the $n$-soliton solution $\omega^{[n]}$ without using the complicated associated Legendre equation. By comparing with the solution given in [2], our method is simpler and systematic.

# 3. Three explicit solutions of the SIdV equation

In this section, we present in detail three explicit solutions generated by the onefold, twofold and threefold DT and the key properties of these solutions, including trajectories, phase shifts after collision and decomposition into separate single kinks.

## 3.1. Solution $u^{[1]}$ generated by onefold Darboux transformation

We set $n = 1$, $\psi = \exp k(x + 4k^2 t)$, $\psi_1 = \cosh k_1(x + 4k_1^2 t)$, then equation (2.5) yields the single-soliton solution of the KdV equation

$$\omega^{[1]} = -2[\ln \cosh k_1(x + 4k_1^2 t)]_{xx} = -2k_1^2 \mathrm{sech}^2 k_1(x + 4k_1^2 t), \tag{3.1}$$

while equation (2.7) implies

$$u^{[1]} = \psi^{[1]}|_{\lambda=0} = -k_1 \tanh k_1(x + 4k_1^2 t). \tag{3.2}$$

The above solution $u^{[1]}$ is a single kink-type soliton that is plotted in figure 1. The line in figure 1$b$ denotes the trajectory of the soliton, which is defined by $x + 4k_1^2 t = 0$. It is easy to find that, for all $t \in R$, $\lim_{x \to +\infty} u^{[1]} = -k_1$ and $\lim_{x \to -\infty} u^{[1]} = k_1$. Note that the solutions given by equations (3.1) and (3.2) are the same as those given in [2], which were obtained by solving the associated Legendre equation.

## 3.2. Solution $u^{[2]}$ generated by the twofold Darboux transformation

We set $n = 2$, and we obtain the two-soliton solution of the KdV equation as

$$\omega^{[2]} = \frac{2(k_1^2 \cosh^2 k_2(x + 4k_2^2 t) + k_2^2 \cosh^2 k_1(x + 4k_1^2 t) - k_1^2)(k_1^2 - k_2^2)}{(k_2 \cosh k_1(x + 4k_1^2 t) \cosh k_2(x + 4k_2^2 t) - k_1 \sinh k_1(x + 4k_1^2 t) \sinh k_2(x + 4k_2^2 t))^2}, \tag{3.3}$$

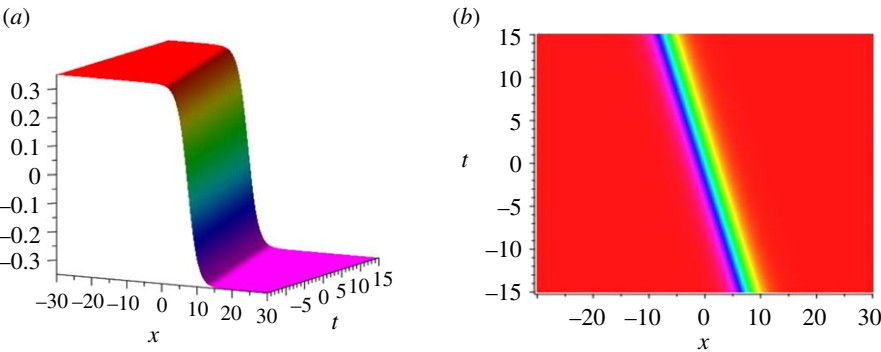

**Figure 1.** (a,b) The dynamical evolution of the single kink-type solution $u^{[1]}$ with $k_1 = 0.35$. The right panel is the density plot of the left panel.

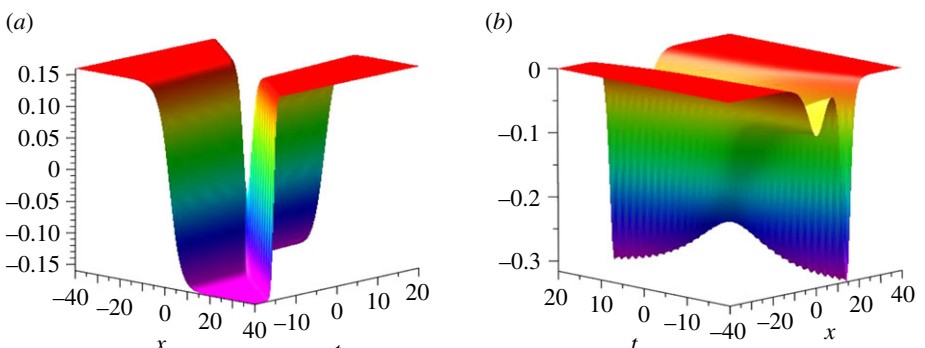

**Figure 2.** The dynamical evolutions of the two-kink solution $u^{[2]}$ (a) and the corresponding two-soliton solution $\omega^{[2]}$ (b) with parameters $k_1 = 0.2$, $k_2 = 0.8$.

and the two-kink solution of the SIdV equation as

$$u^{[2]} = \psi^{[2]}|_{\lambda=0} = \frac{A_1}{B_1},$$ (3.4)

$$A_1 = k_1^2 k_2 \cosh k_2 (x + 4k_2^2 t) \cosh k_1 (x + 4k_1^2 t) - k_2^2 k_1 \sinh k_1 (x + 4k_1^2 t)$$

$$\sinh k_2 (x + 4k_2^2 t),$$

$$B_1 = k_1 \sinh k_1 (x + 4k_1^2 t) \sinh k_2 (x + 4k_2^2 t) - k_2 \cosh k_2 (x + 4k_2^2 t)$$

$$\cosh k_1 (x + 4k_1^2 t),$$

according to equations (2.5) and (2.7). These solutions are plotted in figure 2.

By simple calculations, $u^{[2]}$ is re-formulated as

$$u_a^{[2]} = k_1 k_2 \frac{-\tanh \xi_2 \cosh \xi_0 \sinh \xi_1 + \sinh \xi_0 \cosh \xi_1}{\tanh \xi_2 \sinh \xi_0 \sinh \xi_1 - \cosh \xi_0 \cosh \xi_1}.$$ (3.5)

Here $\xi_i = k_i (x + 4k_i^2 t)$ $(i = 1, 2)$, $\xi_0 = (1/2) \ln((k_1 + k_2)/(k_2 - k_1))$ denotes the phase shift after the interaction of the two kinks. Furthermore, when $\xi_2 \sim \pm\infty$, $u_a^{[2]} \sim -k_1 k_2 \tanh(\xi_0 \mp \xi_1)$. Similarly, $u^{[2]}$ is also re-expressed by

$$u_b^{[2]} = k_1 k_2 \frac{-\tanh \xi_1 \cosh \xi_0 \sinh \xi_2 + \sinh \xi_0 \cosh \xi_2}{\tanh \xi_1 \sinh \xi_0 \sinh \xi_2 - \cosh \xi_0 \cosh \xi_2},$$ (3.6)

which implies that $u_b^{[2]} \sim -k_1 k_2 \tanh(\xi_0 \mp \xi_2)$ when $\xi_1 \sim \pm\infty$. According to the above asymptotic analysis of $u_a^{[2]}$ and $u_b^{[2]}$, it is easy to find $u^{[2]} \sim k_1 k_2$ when $\xi_1$ and $\xi_2$ tend to $\pm\infty$ simultaneously. In other words, the heights of higher and lower asymptotic planes of $u^{[2]}$ are $\pm k_1 k_2$, which is confirmed by figure 2.

We are now in a position to study the decomposition of $u^{[2]}$. As the usual decomposition of a two-soliton solution, we set a trial decomposition of $u^{[2]}$ as a combination of $u_a^{[2]}$ with $\xi_2 \sim -\infty$ and $u_b^{[2]}$

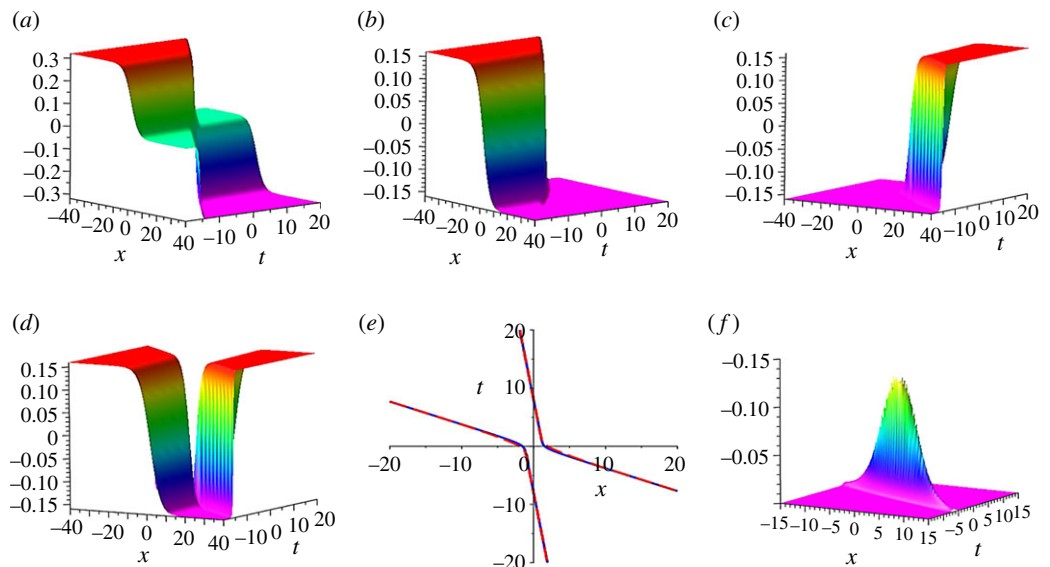

**Figure 3.** The construction of decomposition of $u^{[2]}$ with parameters $k_1 = 0.2$, $k_2 = 0.8$. (a) $u^{[2]}_{\text{trial}}$, (b) $u^{[2]}_{\text{left}}$, (c) $u^{[2]}_{\text{right}}$, (d) $u^{[2]}_{\text{dec}}$ (the decomposition of $u^{[2]}$), (e) the contour lines defined by $u^{[2]} = 0$ (blue, solid) and $u^{[2]}_{\text{dec}} = 0$ (red, dash) and (f) $u^{[2]} - u^{[2]}_{\text{dec}}$.

with $\xi_1 \sim -\infty$, namely

$$u^{[2]}_{\text{trial}} \sim -k_1 k_2 (\tanh(\xi_0 + \xi_2) + \tanh(\xi_0 + \xi_1)),$$

which is plotted in figure 3a. It is clear that figure 3a is a worse approximation of the left part in figure 2a for $u^{[2]}$, because the former has three remarkable differences as compared to the latter: (1) the three plateaux, (2) the height of the top asymptotic plateau and (3) the height of the bottom asymptotic plateau. In order to overcome the inaccuracy of $u^{[2]}_{\text{trial}}$, we introduce $z_1 = -k_1 k_2 (\tanh(\xi_0 + \xi_1) + \tanh(\xi_0 + \xi_2))$ and set

$$u^{[2]}_{\text{left}} = z_1 H(z_1) - k_1 k_2,$$

which is an excellent approximation of the left part in figure 2a for $u^{[2]}$; see figure 3b. Here $H$ is the Heaviside function, $H(z) = \begin{cases} 0, & \text{if } z < 0, \\ 1, & \text{if } z \geq 0. \end{cases}$ Next, we use a combination of $u^{[2]}_a$ with $\xi_2 \sim +\infty$ and $u^{[2]}_b$ with $\xi_1 \sim +\infty$, and we introduce $z_2 = -k_1 k_2 (\tanh(\xi_0 - \xi_1) + \tanh(\xi_0 - \xi_2))$, then

$$u^{[2]}_{\text{right}} = z_2 H(z_2) - k_1 k_2$$

is an excellent approximation of the right part in figure 2a for $u^{[2]}$ (figure 3c). Furthermore, using $u^{[2]}_{\text{left}}$ and $u^{[2]}_{\text{right}}$, we provide a decomposition of $u^{[2]}$, namely

$$u^{[2]}_{\text{dec}} = \begin{cases} z_2 H(z_2) - k_1 k_2, & \text{if } \xi_1, \xi_2 \gg 0, \\ z_1 H(z_1) - k_1 k_2, & \text{if } \xi_1, \xi_2 \ll 0. \end{cases} \tag{3.7}$$

Thus, we get an excellent approximate decomposition of $u^{[2]}$, which is shown in figure 3d,e. Here we plot $u^{[2]}_{\text{dec}}$ in figure 3d, and in figure 3e we plot the two contour lines defined by $u^{[2]} = 0$ (blue, solid) and $u^{[2]}_{\text{dec}} = 0$ (red, dash). However figure 3f shows a remarkable discrepancy $u^{[2]} - u^{[2]}_{\text{dec}}$ in a small region of strong interaction of the two kinks, around $t = 0$.

## 3.3. Solution $u^{[3]}$ generated by the threefold Darboux transformation

Setting $n = 3$ in equations (2.5) and (2.7), a three-soliton solution $\omega^{[3]}$ of the KdV equation and a three-kink solution $u^{[3]}$ of the SIdV equation can be written out explicitly. These two types of soliton solutions are plotted in figure 4. Due to the lack of space, we only provide here the explicit formula of a three-kink solution, namely

$$u^{[3]} = \psi^{[3]}|_{\lambda=0} = \frac{A_2}{B_2}. \tag{3.8}$$

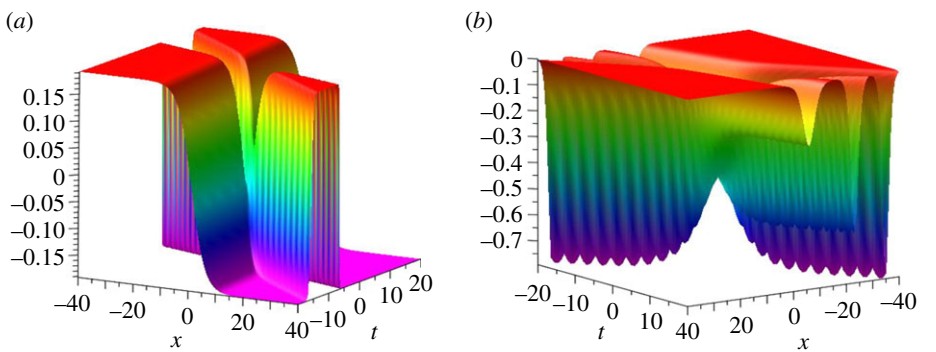

**Figure 4.** The dynamical evolutions of the three-kink solution $u^{[3]}$ (a) and of the corresponding three-soliton solution $\omega^{[3]}$ (b) with parameters $k_1 = 0.2$, $k_2 = 0.8$, $k_3 = 1.2$.

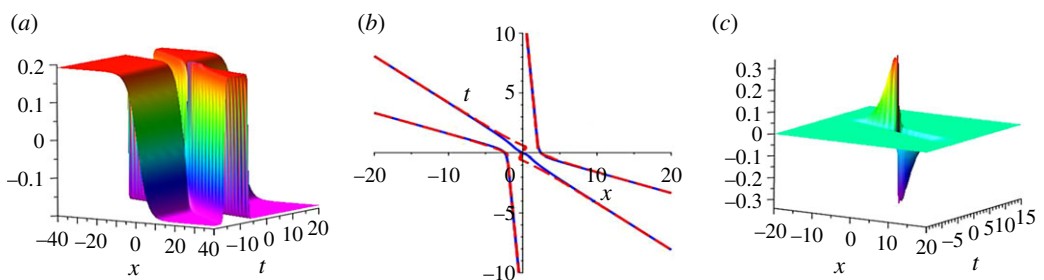

**Figure 5.** The decomposition of $u^{[3]}$ with parameters $k_1 = 0.2$, $k_2 = 0.8$ and $k_3 = 1.2$. (a) $u_{\text{dec}}^{[3]}$ (the decomposition of $u^{[3]}$), (b) the contour lines defined by $u^{[3]} = 0$ (blue, solid) and $u_{\text{dec}}^{[3]} = 0$ (red, dash) and (c) $u^{[3]} - u_{\text{dec}}^{[3]}$.

Here $\xi_i = k_i(x + 4k_i^2 t)$, $i = 1, 2, 3$,

$$A_2 = k_1 k_2 k_3[- k_1(k_2 - k_3)(k_2 + k_3) \cosh \xi_1 \cosh \xi_2 \sinh \xi_3 - k_3(k_1 - k_2)(k_1 + k_2)$$
$$\sinh \xi_1 \cosh \xi_2 \cosh \xi_3 + k_2(k_1 - k_3)(k_1 + k_3) \sinh \xi_1 \sinh \xi_2 \sinh \xi_3]$$

and

$$B_2 = k_1 k_2 k_3[- k_2(k_1 - k_3)(k_1 + k_3) \cosh \xi_1 \cosh \xi_2 \cosh \xi_3 + k_1(k_2 - k_3)(k_2 + k_3)$$
$$\sinh \xi_1 \sinh \xi_2 \cosh \xi_3 + k_3(k_1 - k_2)(k_1 + k_2) \cosh \xi_1 \sinh \xi_2 \sinh \xi_3].$$

By a similar calculation as done for $u^{[2]}$, the solution $u^{[3]}$ is decomposed approximately into three separate kinks in the following form, namely:

$$u_{\text{dec}}^{[3]} = \begin{cases} k_1 k_2 k_3[\tanh(\eta_1 - \xi_1) + \tanh(\eta_3 - \xi_3) - (\tanh(\eta_2 - \xi_2) \\ \quad H(y) + \tanh(\eta_2 + \xi_2)H(-y))]H(-z), & \xi_1, \xi_2, \xi_3 \gg 0, \\ k_1 k_2 k_3[\tanh(\eta_1 + \xi_1) + \tanh(\eta_3 + \xi_3) - (\tanh(\eta_2 - \xi_2) \\ \quad H(y)) + \tanh(\eta_2 + \xi_2)H(-y))]H(z), & \xi_1, \xi_2, \xi_3 \ll 0. \end{cases} \quad (3.9)$$

Here $\eta_1 = (1/2)\ln(((k_2(k_3^2 - k_1^2) - k_3(k_2^2 - k_1^2) + k_1(k_3^2 - k_2^2))/(k_2(k_3^2 - k_1^2) - k_3(k_2^2 - k_1^2) - k_1(k_3^2 - k_2^2))))$, $\eta_2 = (1/2)\ln(((k_2(k_3^2 - k_1^2) + k_3(k_2^2 - k_1^2) + k_1(k_3^2 - k_2^2))/(k_2(k_3^2 - k_1^2) - k_3(k_2^2 - k_1^2) - k_1(k_3^2 - k_2^2))))$, $\eta_3 = (1/2)\ln(((k_2(k_3^2 - k_1^2) + k_3(k_2^2 - k_1^2) - k_1(k_3^2 - k_2^2))/(k_2(k_3^2 - k_1^2) - k_3(k_2^2 - k_1^2) - k_1(k_3^2 - k_2^2))))$, $z = k_1 k_2 k_3(\tanh(\eta_3 - \xi_3)H(y) - \tanh(\eta_3 + \xi_3)H(-y))$ and $y = x - t$. Moreover, in figure 5a we plot $u_{\text{dec}}^{[3]}$, and in figure 5b we plot the corresponding three contour lines defined by $u^{[3]} = 0$ (blue, solid) and $u_{\text{dec}}^{[3]} = 0$ (red, dash), which show an excellent agreement between $u^{[3]}$ and $u_{\text{dec}}^{[3]}$. However, figure 5c shows a remarkable discrepancy $u^{[3]} - u_{\text{dec}}^{[3]}$ in a small region of strong interaction of the three kinks, around $t = 0$.

## 4. Conclusion

In this paper, the order-$n$ kink-type solution $u^{[n]}$ of the SIdV equation (1.4), which is associated with an $n$-soliton solution of the KdV equation, is constructed by using the $n$-fold DT from zero 'seed' solution. The trajectories, the phase shifts, and the decomposition of the first three kink-type solutions $u^{[n]}$ ($n = 1, 2,$

3) are studied in detail. A crucial relationship is $u = \psi|_{\lambda=0}$, so we can use the DT to construct the solution $u^{[n]}$ without using the solution of the associated Legendre equation as was done in [2]. By comparing with the results reported in [2], we believe that our method presented here is simpler and systematic. Moreover, we mention that the SIdV equation is also used to describe and control the revolution of surfaces [7,8], thus it is an interesting issue to get explicitly the surfaces associated with the order-$n$ kink soliton $u^{[n]}$. These results will be reported elsewhere.

Data accessibility. This article has no additional data.

Authors' contributions. G.F.Z. and J.S.H. carried out the solution and drafted the initial manuscript; L.H.W. participated in numerical analysis of solution; D.M. provided physical consideration and helped draft the manuscript. All authors gave final approval for publication.

Competing interests. We have no competing interests.

Funding. This work is supported by the NSF of China under grant no. 11671219, the Natural Science Foundation of Zhejiang Province under grant nos. LZ19A010001 and LSY19A010002, and the K.C. Wong Magna Fund in Ningbo University.

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
