## [Reviewer comments · Royal Society Open Science]

Review History

RSOS-191040.R0 (Original submission)

Review form: Reviewer 1

Is the manuscript scientifically sound in its present form?

Yes

Are the interpretations and conclusions justified by the results?

Yes

Is the language acceptable?

Yes

Do you have any ethical concerns with this paper?

No

Have you any concerns about statistical analyses in this paper?

No

Recommendation?

Reject

Comments to the Author(s)

There are many well-known ways to generate exact solutions of a nonlinear integrable equation, e.g. inverse scattering techniques, dressing method, Hirota bilinear method, Bäcklund transformations, Darboux transformations, traveling wave ansatz, etc. In the present paper, the authors start with a seed solution and generate some new solutions for Eq. (1.1) using Darboux transformations. It's a straightforward analysis and the techniques are standard modulo minor modifications. The paper is not interesting or novel enough for publication in a high-level journal like RSOS.

Review form: Reviewer 2**Is the manuscript scientifically sound in its present form?**

Yes

Are the interpretations and conclusions justified by the results?

Yes

Is the language acceptable?

Yes

Do you have any ethical concerns with this paper?

No

Have you any concerns about statistical analyses in this paper?

No

Recommendation?

Accept as is

Comments to the Author(s)

In this revised version, some corrections have been made. I think the paper is suitable for publication in RSOS.

Decision letter (RSOS-191040.R0)

18-Jul-2019

Dear Dr He

On behalf of the Editors, I am pleased to inform you that your Manuscript RSOS-191040 entitled "Kink-type solutions of the SIdV equation and their properties" has been accepted for publication in Royal Society Open Science subject to minor revision in accordance with the referee suggestions. Please find the referees' comments at the end of this email.

The reviewers and handling editors have recommended publication, but also suggest some minor revisions to your manuscript. Therefore, I invite you to respond to the comments and revise your manuscript.

- Ethics statement

- Data accessibility

If you wish to submit your supporting data or code to Dryad (<http://datadryad.org/>), or modify your current submission to dryad, please use the following link:
<http://datadryad.org/submit?journalID=RSOS&manu=RSOS-191040>

- Competing interests

- Authors' contributions

- Acknowledgements

- Funding statement

Because the schedule for publication is very tight, it is a condition of publication that you submit the revised version of your manuscript before 27-Jul-2019. Please note that the revision deadline will expire at 00.00am on this date. If you do not think you will be able to meet this date please let me know immediately.

Kind regards,
Lianne Parkhouse
Editorial Coordinator
Royal Society Open Science
openscience@royalsociety.org

on behalf of Professor Takashi Suzuki (Associate Editor) and Mark Chaplain (Subject Editor)
openscience@royalsociety.org

Reviewer comments to Author:
Reviewer: 1

Comments to the Author(s)

There are many well-known ways to generate exact solutions of a nonlinear integrable equation, e.g. inverse scattering techniques, dressing method, Hirota bilinear method, Bäcklund transformations, Darboux transformations, traveling wave ansatz, etc. In the present paper, the authors start with a seed solution and generate some new solutions for Eq. (1.1) using Darboux transformations. It's a straightforward analysis and the techniques are standard modulo minor modifications. The paper is not interesting or novel enough for publication in a high-level journal like RSOS.

Reviewer: 2

Comments to the Author(s)

In this revised version, some corrections have been made. I think the paper is suitable for publication in RSOS.

Staff note: Please see the attached PDF for reviewer comments.

Author's Response to Decision Letter for (RSOS-191040.R0)

See Appendix A.

Decision letter (RSOS-191040.R1)

25-Jul-2019

Dear Dr He,

I am pleased to inform you that your manuscript entitled "Kink-type solutions of the SIV equation and their properties" is now accepted for publication in Royal Society Open Science.

on behalf of Professor Takashi Suzuki (Associate Editor) and Mark Chaplain (Subject Editor)
openscience@royalsociety.org

Appendix A

Dear Editor,

Thank you very much for your acceptance of our submission in publication in RSOS.

As you know, our submission is transferred from PRSA, and we have done correction according to the three reports of PRSA:

Reviewer 1: the report is the same as the report of PRSA except the change of the journal title (PRSA to RSOS).

Reviewer 2: The reviewer has read our revised version and recommended the acceptance of our paper in RSOS.

Reviewer 3: The report is the same as the report of PRSA without any change including the title of journal.

So, we are just replacing PRSA's paper template with RSOS's template.

Thank you again.

Jingsong He